# Customized versus Standard Epithelium Profiles in Transepithelial Photorefractive Keratectomy

**Diego de Ortueta** [1,*] **, Dennis von Rüden** [1] **and Samuel Arba-Mosquera** [2]

1   AURELIOS Augenzentrum Recklinghausen, 45657 Recklinghausen, Germany; dennis.von.rueden@augenzentrum.org
2   SCHWIND Eye-Tech-Solutions, 63801 Kleinostheim, Germany; Samuel.arba.mosquera@eye-tech.net
*   Correspondence: diego.de.ortueta@augenzentrum.org

**Abstract:** Transepithelial photorefractive keratectomy (TransPRK) is an established surface ablation technique used to correct refractive errors. Using anterior segment optical coherence (AS-OCT), it is now possible to measure the epithelium thickness and input these data into the laser platform. In this study, we explore whether better results were obtained in this way. To this end, we retrospectively analyze the results from a low-myopia group treated with a customized epithelium thickness, as measured using AS-OCT, and compare them with the results from a group treated with an optimized standard epithelium thickness. The customized epithelium profile group contains more eyes with vision better than 20/20, and more eyes in this group gain one line of corrected distance visual acuity (CDVA). In conclusion, with the customized epithelium thickness, we obtain superior results using TransPRK in low-myopia corrections.

**Keywords:** TransPRK; PRK; transepithelial photorefractive keratectomy; anterior segment optical coherence tomography; MS-39; myopia; epithelium; epithelium thickness

## 1. Introduction

Transepithelial photorefractive keratectomy (TransPRK) is a one-step surface ablation technique in which the epithelium and stroma are ablated in order to correct refractive errors. Using the AMARIS laser platform (SCHWIND eye-tech-solutions), the epithelium is ablated with phototherapeutic keratectomy (PTK, and the stroma, as well as with aspherical photorefractive keratectomy (PRK). The PTK and PRK profiles, together, provide the TransPRK profile. The software pulses for the correction of a refractive error are given in reverse mode, so the PRK profile is used first, and then, the epithelium PTK pulses are dispatched to the cornea [1].

We recently published the refractive and visual outcomes after TransPRK treatment for low-myopia (2 diopters (D) or less) [2] corrections, using a larger optical zone and a thicker epithelium than those proposed by the standard software. One of the main difficulties with low-myopia corrections with TransPRK is that, when the epithelium is thicker than the precalculated value, the ablation zone becomes smaller and, sometimes, no correction occurs [3].

With anterior segment coherence tomography (AS-OCT), we can measure the epithelial thickness with a high repeatability and reproducibility [4,5]. With the MS-39 (Costruzione Strumenti Oftalmici, Florence, Italy), we can achieve non-contact corneal imaging with spectral-domain anterior segment OCT, combined with Placido-based topography, which can detect different corneal layers, separately, with a very high resolution and a wide-field epithelial thickness map [6]. The purpose of this study was to analyze the results of TransPRK in low-myopia eyes treated with a customized epithelium thickness and compare them with the results from a group of eyes treated with an optimized standard corneal epithelium thickness. An optimized standard corneal epithelium thickness denotes

an epithelium of 60 microns rather than the 55 microns proposed by the software, and it enlarges the optical zone (OZ) at least 0.2 mm larger than the proposed by the software.

## 2. Materials and Methods

We retrospectively analyzed two consecutive groups of eyes treated with TransPRK for low myopia (2 diopters (D) or less). From January 2020, we routinely used the MS-39 (Costruzione Strumenti Oftalmici, Florence, Italy) to measure the central epithelium thickness and used the mean thickness data for the laser platform input. We evaluated a group of consecutive eyes until February 2021, with a follow up of at least 4 months. This group was compared with the same number of eyes, treated with the optimized standard epithelium thickness, between January 2017 and December 2019.

The study was conducted in accordance with the Declaration of Helsinki. The institutional review board approved this retrospective evaluation. Proper informed consent was obtained from each patient, and the data were de-identified for clinical data calculation and publication. Patients were enrolled in the study if they had a preoperative corrected distance visual acuity (CDVA) of 20/32 or better according to the standardized Snellen Charts from the international standardization organization (ISO), had had stable refraction for 1 year before the study, and had discontinued contact lens use for at least 2 weeks before the preoperative evaluation.

### 2.1. Clinical Evaluation

Preoperatively, all the patients underwent a complete ophthalmological examination, with the determination of refractive defects under manifest and cycloplegic conditions, measuring the uncorrected distance visual acuity (UDVA) and best-corrected distance visual acuity (CDVA), performing pupillometry, performing corneal topography and Scheimpflug corneal pachymetry with a SIRIUS topo-tomographer (Costruzione Strumenti Oftalmici, Florence, Italy), performing aberrometry with a Peramis (SCHWIND eye-tech-solutions, Kleinostheim, Germany), performing AS-OCT with a MS-39 (Costruzione Strumenti Oftalmici, Florence, Italy), and performing a fundus evaluation with a dilated pupil. One day, postoperatively, the UDVA was measured, and the patient had a slit-lamp examination of the anterior segment. The same examinations as those performed preoperatively (except the dilated fundoscopy unless warranted) were performed at 1 week, 1 month, and 4 months.

### 2.2. Treatment Plan

As previously described [2], we used surface ablation TransPRK with an aberration-free ablation pattern. The treatment was planned with the ORK-CAM planning module. The profile was aspheric, and we used Smart Pulse Technology [7].

### 2.3. Surgical Technique

This retrospective cohort study was based on two consecutive case series of patients treated by a single surgeon (DdO), in which TransPRK was used to correct myopic astigmatism, at Aurelios Augenlaserzentrum, Recklinghausen, Germany.

The sphere and cylinder values based on the manifest refraction were entered into the laser with nomogram adjustments, based on data from previously operated eyes provided by the Datagraph-med software (Wendelstein, Germany). Keratometry data at 3 mm were also used to compensate for the geometry of the eye [8].

We used the AMARIS laser platform (SCHWIND eye-tech-solutions, Kleinostheim, Germany) with a 1050 Hz infrared eye tracker with simultaneous limbus, pupil, and torsion tracking, centering the ablation on the corneal vertex [9,10].

After surgery, a soft bandage contact lens (Air Optix Night & Day, base curve 8.4) was applied for 4 days. The patients took dexamethasone eye drops without preservatives (Dexa Edo, Bausch + Lomb, Berlin, Germany) and orfloxacin eye drops without preservatives (Floxal Edo, Bausch + Lomb, Berlin, Germany) four times a day for 1 week, fluormetholone eye drops (Fluoropos Ursapharm GmbH, Saarbrücken, Germany) three

times per day for another 6 weeks, and preservative-free lubricants for 2 months and beyond, as necessary.

In Group A, with the standardized optimized epithelial thickness profile, we input the central epithelium thickness at 60 and 70 microns at a 4 mm radial distance, as described previously [2]. In Group B, with the customized epithelium thickness profile, the input was the mean epithelium thickness, as measured with the AS-OCT, and the 4 mm input was automatically increased, by the software, to 10 microns.

Readers are referred to ref. [3], in which the implications and the rationale of the decision is fully disclosed. Table 1 summarizes the differences between the different ablation profiles for the epithelium:

**Table 1.** Summarizes the different ablation profiles of the epithelium for TransPRK.

| Epithelial Profile | Central Thickness | Peripheral Thickness @4 mm Radial Distance | Remarks |
|---|---|---|---|
| Standard | 55 μm | 65 μm | Population based epithelium covering >75% of the normal population |
| Optimized (Standard + 5 μm) | 60 μm | 70 μm | Optimized population based epithelium covering >95% of the normal population |
| Customized (as measured by OCT) | 40–75 μm (as measured centrally using MS-39) | Central thickness + 10 μm | According to Impact of the Reference Point for Epithelial Thickness Measurements. Arba-Mosquera S, Awwad ST. J Refract Surg. 1 March 2020; 36(3):200–207 |

*2.4. Excimer Laser*

The laser ablation algorithm has been described previously [2]. In brief, a flying-spot delivery system that operates at 1050 Hz with a super-Gaussian beam profile of 0.54 mm full width-half maximum [11] is used. It works with an inverted sequentialization of the epithelium profile and aspherical correction without breaks. Depending on the planned refractive correction, approximately 80% of the corneal ablation was performed with a high fluence level (~440 mJ/cm$^2$) and 20%, with a low fluence level (~300 mJ/cm$^2$) for the fine correction and to smooth the ablated area [7]. An aspiration system with laminar flow dynamics was incorporated to reduce debris and heat build-up.

*2.5. Data Analysis*

The refractive and visual outcomes were analyzed using the Excel software (Microsoft Corp.). The logMAR visual acuities were converted to Snellen acuities for data reporting using the Visual Acuity Conversion Chart of the Journal of Cataract & Refractive Surgery. The uncorrected and corrected visual acuity, spherical equivalent refraction, and refractive astigmatism were evaluated. A *p* value of less than 0.05 was considered statistically significant. Data up to 4 months post-operation are reported herein. The normality of the samples was assessed using the back-of-the-envelope and the quantile–quantile methods. The intergroup comparisons were assessed using unpaired Student's *t*-tests, whereas preoperative to postoperative changes were assessed using paired Student's *t*-tests. We used both eyes from the same subject, yet, for the statistical analyses, the *p*-values were calculated considering the number of patients and not the number of eyes.

**3. Results**

*3.1. Demographics*

Dempgraphic data and data analysis are summarized in Table 2.

**Table 2.** Summarizes the values used for both groups, including age, astigmatism magnitude, and CDVA.

|  | Standard Epithelium X ± SD Range | | Customized Epithelium X ± SD Range | | *p*-Value between Groups |
|---|---|---|---|---|---|
| Number of eyes | 58 | | 58 | | — |
| Age (years) | 35 ± 11 | 18 to 64 | 32 ± 9 | 18 to 56 | 0.1 |
| UDVA (Snellen) | 20/80 ± 11 | 20/20 to 20/400 | 20/80 ± 9 | 20/20 to 20/400 | 0.4 |
| CDVA (Snellen) | 20/18 ± 4 | 20/12 to 20/32 | 20/18 ± 3 | 20/12 to 20/32 | 0.5 |
| Spherical equivalent (D) | −1.41 ± 0.43 | −2.0 to −0.5 | −1.43 ± 0.36 | −2 to −0.75 | 0.4 |
| Astigmatism (D) | 0.8 ± 0.69 | 0 to 2.5 | 0.83 ± 0.70 | 0 to 3 | 0.4 |
| Central corneal thickness (μm) | 551 ± 35 | 442 to 666 | 553 ± 26 | 467 to 657 | 0.4 |
| Optical zone (mm) | 7.1 ± 0.2 | 6.7 to 7.7 | 6.8 ± 0.2 | 6.3 to 7.4 | <0.0001 |
| Total ablation zone (mm) | 8.2 ± 0.3 | 7.5 to 9.5 | 7.9 ± 0.3 | 7.4 to 9.0 | <0.0001 |
| Total ablation depth (μm) | 90 ± 9 | 67 to 142 | 86 ± 9 | 88 to 175 | 0.01 |

### 3.2. Epithelium Thickness

In the group of eyes treated with the customized epithelium profile, we used the corneal vertex measurement data with the AS-OCT. The epithelium thickness at the vertex entered into the SCHWIND AMARIS was the mean of three measurements of each eye. The median and mean epithelium thickness, in our treated population with the customized epithelium thickness profile, was 54.5 microns, with a standard deviation of 4.14 microns and a range between 49 and 72 microns.

### 3.3. Predictability

The predictability of the treatments was the same in the two groups (Figures 1–4). The scattergram of the achieved vs. attempted correction was very similar for both groups (Figure 1). The deviation from the intended target was −0.03 ± 0.31 D in the standard epithelium group and +0.09 ± 0.29 D in the customized epithelium group (*p* < 0.0001); the postoperative SEQ was −0.01 ± 0.30 D in the standard epithelium group and −0.08 ± 0.28 D in the customized epithelium group (*p* < 0.0001) (Figure 2). The postoperative refractive astigmatism was 0.28 ± 0.09 D in both groups (*p* = 0.5) (Figure 3). The SEQ did not change in a relevant manner, between 1 M and 4 M, in either group (Figure 4).

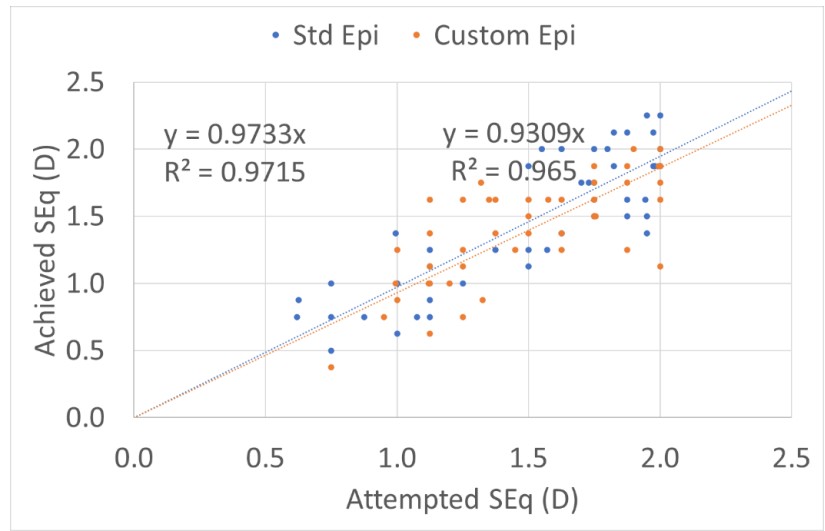

**Figure 1.** Predictability of the achieved vs. attempted spherical equivalent (SEQ) for both low-myopia groups. The customized epithelium group exhibited a small undercorrection. The regression analysis and R coefficient of determination, in both groups, were near to 1. The predictability of the treatments was similar in the two groups. The scattergram of the achieved vs. attempted correction was very similar for both groups.

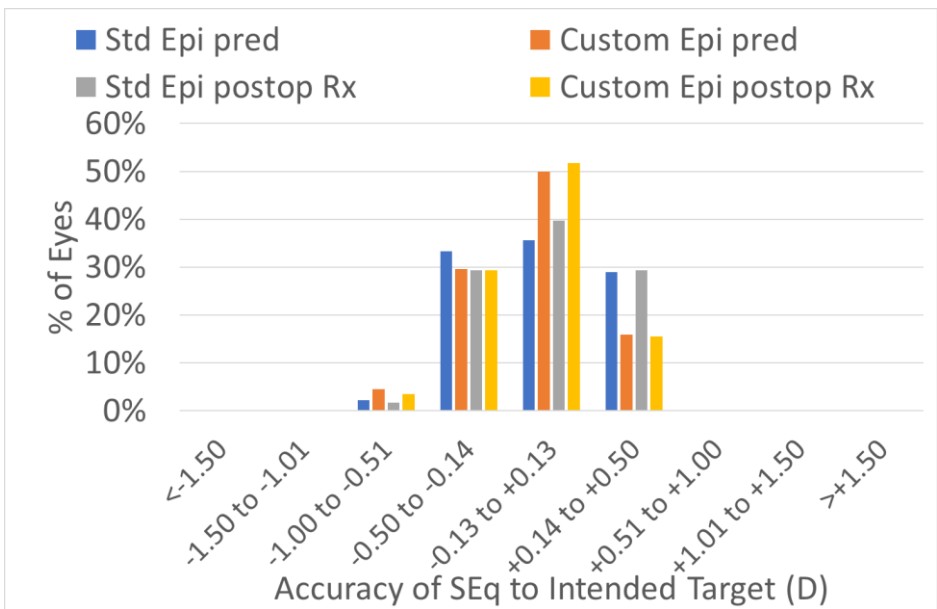

**Figure 2.** Accuracy expressed in SEQ deviation from the achieved vs. attempted groups. The predictability of the treatments was similar in the customized vs. standard groups. In 100% of the eyes in the low-myopia group, with the standard epithelium and customized epithelium profiles, the SEQ was within 1.0 D of the intended target. In 95% of the eyes with the standard epithelium profile group and 94% of the eyes in the customized epithelium profile group, the SEQ was within 0.5 D of the intended target.

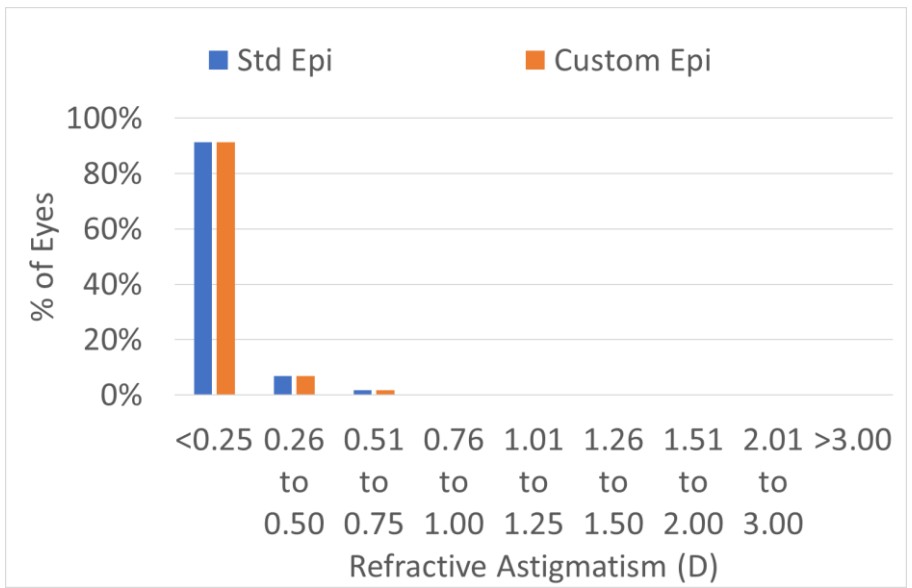

**Figure 3.** Postoperative refractive astigmatism. Postoperative refractive astigmatism was 0.28 ± 0.09 D in both groups (*p* = 0.5). The predictability of the treatments was the same in both groups. In 100% of the eyes, the postoperative refractive astigmatism was within 0.5 D of the intended target.

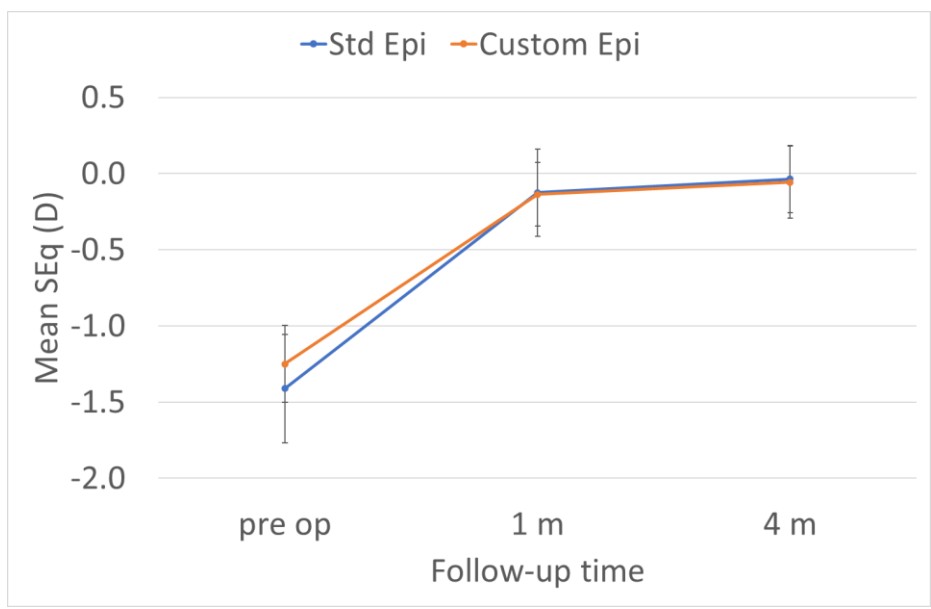

**Figure 4.** The SEQ did not change in a relevant manner between 1 M and 4 M in either group.

*3.4. Efficacy*

The efficacy of the treatments was significantly improved in the customized epithelium profile group (Figure 5). The preoperative CDVA was 20/18 ± 3 in the standard epithelium profile group and 20/18 ± 4 in the customized epithelium profile group ($p$ = 0.1); the postoperative UDVA was 20/20 ± 4 in the standard epithelium profile group and 20/18 ± 4 in the customized epithelium profile group; the postoperative UDVA was three optotypes better ($p$ = 0.001). The postoperative UDVA was the same as the preoperative CDVA, i.e., 0 ± 0.7 lines, for the customized epithelium profile group; the UDVA was −0.4 ± 0.7 lines worse than the preoperative CDVA in the standard epithelium profile group ($p$ = 0.3) (Figure 6).

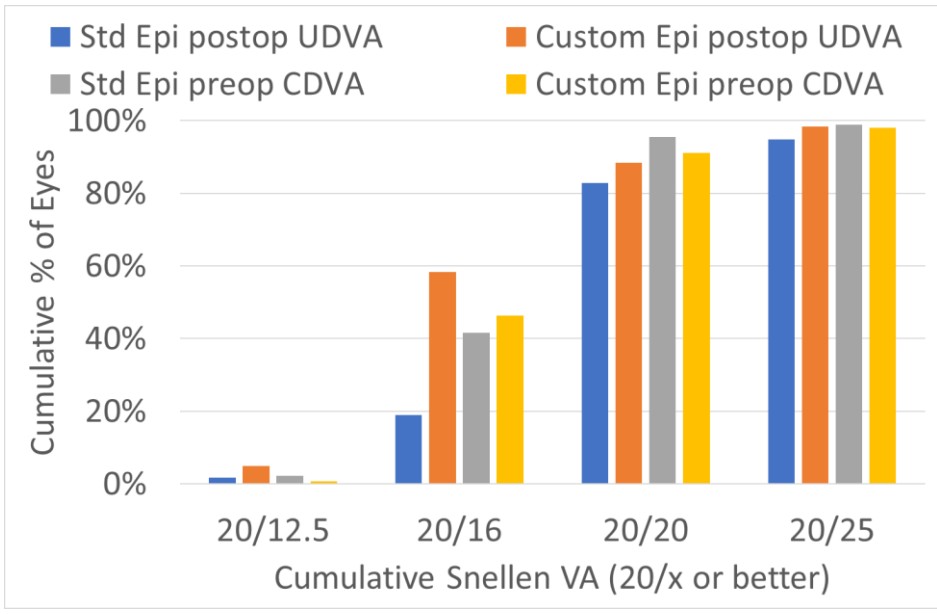

**Figure 5.** Efficacy of the preoperatively corrected distance visual acuity (CDVA) and the postoperatively uncorrected distance visual acuity (UDVA). The customized epithelium profile group contained more eyes, with a UDVA of 20/16 or better, postoperatively.

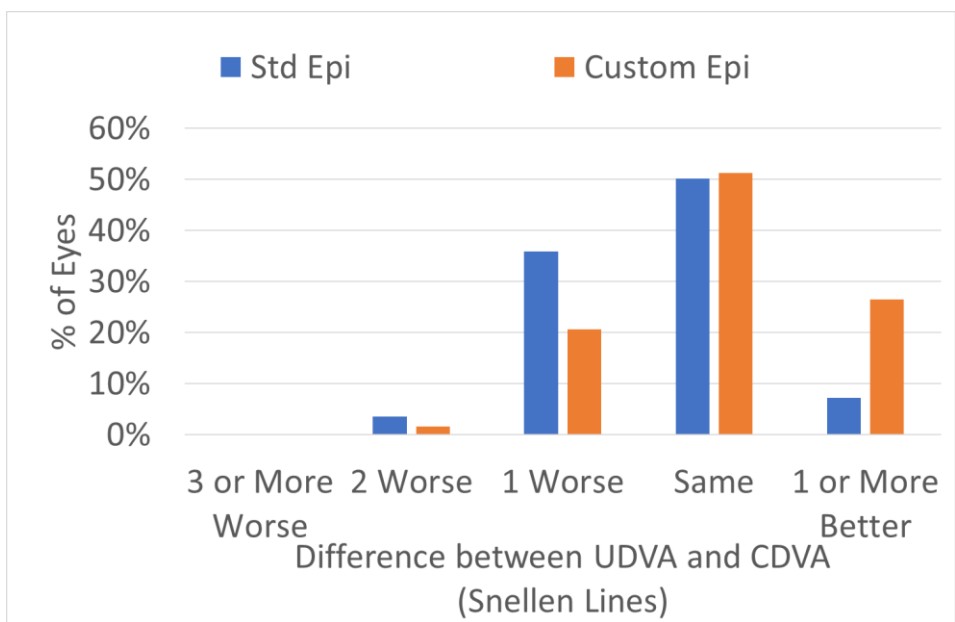

**Figure 6.** Efficacy expressed in Snellen lines. The post-operative UDVA was the same as the preoperative CDVA, i.e., $0 \pm 0.7$ lines, for the customized epithelium group; the UDVA was $-0.4 \pm 0.7$ lines worse than the preoperative CDVA in the standard epithelium profile group ($p = 0.3$).

*3.5. Safety*

In terms of safety, no eye lost two or more Snellen acuity lines (Figure 7). The postoperative CDVA was the same as the preoperative CDVA in the standard epithelium profile group ($p = 0.1$). In the customized epithelium profile group, the CDVA was two optotypes better than the preoperative CDVA ($p < 0.0001$). Moreover, the difference was significant ($p < 0.0001$), with a better CDVA for the customized epithelium profile group.

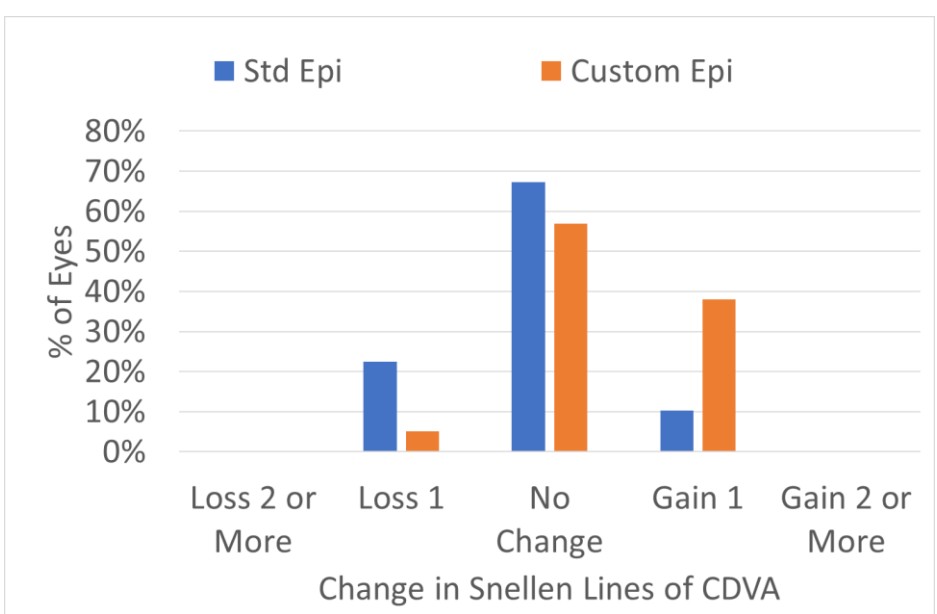

**Figure 7.** Safety change for preoperative CDVA vs. postoperative Snellen lines. No eye lost two or more CDVA lines in either group. More eyes in the customized epithelium profile group gained one line of CDVA.

*3.6. Adverse Events*

No adverse events or complications were observed intra- or post-operatively.

## 4. Discussion

Transepithelial approaches modify the cornea through the epithelium. The ablation rate is higher in the epithelium than in the stroma, and it increases with stromal depth [12]. The difficulty lies in performing corrections with TransPRK for low-myopia corrections, when the epithelial thickness cannot be measured, because if the epithelium is thicker than the standard given by the software (55 microns) at the center, the OZ may be smaller than the precalculated value, which can result in undercorrection [3]. Therefore, in a previous paper, we proposed [2] that, in cases in which the epithelial thickness cannot be measured, the OZ should be enlarged by at least 0.2 mm, and the central epithelium thickness input should be 60 microns. These adjustments affect more tissue, as the laser penetrates deeper than necessary, but they can be used to safely correct low myopia with TransPRK. Optical coherence tomography (OCT) is a non-contact procedure that is based on the principle of interferometry [13]. Using anterior segment optical coherence tomography (AS-OCT), it is possible to measure the epithelial thickness in a reproducible way [5] and, thus, to input the mean epithelial thickness in the laser system. AS-OCT provides an accurate technique with which to measure corneal surfaces, obtain information concerning the epithelial thickness and corneal thickness, and map the epithelium and stroma [14]. There are several AS-OCT instruments on the market that can provide good reproducibility and reliability [15]. We utilized the MS-39 (Costruzione Strumenti Oftalmici, Florence, Italy), which combines AS-OCT with Placido Rings technology. The spectral domain AS-OCT has a 3.5 micron axial resolution for tissue, with the ability to calculate the epithelial and stromal thickness maps over an 8 mm diameter [16]. Several studies [15,17] show that AS-OCT, using the MS-39, provides good reproducibility and repeatability for measurements of corneal thickness.

Using various methods: with OCT, the group of Sin et al. found that the epithelial thickness of normal eyes was 52 +/− 3 mm, with 5th and 95th percentiles for the central epithelial thickness of 48 and 57 microns, respectively. Eckard et al. [18] used confocal laser scanning and reported a mean thickness of 54 +/− 7 µm centrally and 61.5 µm peripherally. Reinstein et al. [19] used high-frequency digital ultrasound techniques and reported a mean epithelial thickness at the corneal vertex of 53.4 +/− 4.6 microns, and the average epithelial thickness map showed that the corneal epithelium was significantly thicker inferiorly rather than superiorly (5.9 µm at the 3.0 mm radius) and significantly thicker nasally rather than temporally (1.3 µm at the 3.0 mm radius). The thinnest point of the epithelium was, on average, located 0.33 mm temporally and 0.90 mm superiorly, with reference to the corneal vertex.

Using a standard epithelial ablation algorithm can, in certain eyes or corneal regions, lead to more stromal ablation than necessary, regardless of the topometry of the epithelial layer. In other eyes (or corneal regions), less epithelium than necessary can be ablated, and a certain amount of the ablation applied to the stroma can ablate the remaining epithelium [3]. We previously obtained good results using TransPRK for low-myopia corrections with an increased standard epithelium of 60 microns and a larger OZ than proposed by the software. Customizing the epithelium thickness for each patient, using AS-OCT, allows one to use the proposed OZ, even in small corrections with TransPRK. In the customized epithelium profile group in our study, the mean central corneal thickness was 54.5, ranging up to 72 microns (very much comparable to the normal epithelia previously reported [17–19]). In particular, eyes with a thicker epithelium can benefit from the correct input into the laser platform, which helps to avoid potential undercorrections and a smaller OZ.

For group A, there was no systematic data on preoperative corneal epithelium thickness. This group was performed before the MS-39 OCT (with the ability to provide epithelial maps) was available at the clinic. We had previous experiences (but not systematic data collections) of using the Optovue OCT to measure the central epithelium in specific patients. We acknowledge this as a limitation since, strictly speaking, we cannot exclude differences between the corneal epithelium thicknesses for the two groups prior to surgery. Yet, at the light of the results in group A and accepting that we were treating normal populations, we "believe" that, even if differences between the corneal epithelium

thickness for the two groups prior to surgery may have existed, this would reach unlikely any meaningful clinical relevance.

For the same reasons, we consider that the report and analysis of postoperative epithelial thickness, which would only be possible for group B, does not add further value to this comparison. The specific analysis of the epithelial remodeling after TransPRK warrants further study.

The accuracy was good for both groups. The safety and efficacy were better in the customized epithelium profile group, with more eyes with a UDVA of more than 20/20 and more eyes gaining one line of CDVA post-operatively.

Jun et al. found that using a standard epithelium thickness for TransPRK corrected fewer astigmatisms in eyes with a thicker epithelium [20]. We did not note any differences in the astigmatism corrections between the groups. This may be because we did not change the center to the periphery at a 4 mm gradient of 10 microns for the thicker epitheliums. Moreover, we exported an epithelial map because, to date, there is no export software for the laser platform. The ability to export epithelial maps could potentially improve TransPRK.

Limitations of this work include the use of both eyes from the same patients (although statistical analyses have been performed based on the number of patients and not the number of eyes), the follow-up time was short (the overall visual recovery is long for (transepithelial) PRK, and 4 months might not be enough to achieve final stability; although the interest in this work was the comparison between both groups at the 4-month follow-up and the final stability); it is a retrospective study, and we are not able to provide corneal epithelium thickness for group A (there could be potential statistically significant difference between the two groups regarding the corneal epithelium thickness, yet it is unlikely to reach any meaningful clinical relevance).

## 5. Conclusions

In this study, we explored whether better results were obtained as a result of changing the central epithelium thickness for each eye. The results from the standard epithelium profile group were good; however, the results from the customized epithelium profile group were superior when using TransPRK in low-myopia corrections.

**Author Contributions:** Conceptualization, D.d.O. and S.A.-M.; methodology, D.d.O.; validation, D.d.O. and S.A.-M.; formal analysis, D.d.O., D.v.R. and S.A.-M.; data curation, D.v.R.; writing—original draft preparation, D.d.O.; writing—review and editing, D.d.O. and S.A.-M.; supervision, D.d.O. All authors have read and agreed to the published version of the manuscript.

**Funding:** This research received no external funding.

**Institutional Review Board Statement:** The study was conducted according to the guidelines of the Declaration of Helsinki, and approved by the Institutional Review Board as a retrospective Study.

**Informed Consent Statement:** Informed consent was obtained from all subjects involved in the study.

**Data Availability Statement:** All data were fully anonymized and are available upon request.

**Conflicts of Interest:** D.d.O. is a consultant for SCHWIND eye-tech-solutions GmbH, S.A.-M. is an employee for SCHWIND eye-tech-solutions, and D.v.R. has no conflict of interest.

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
