# Peer review of "Customized versus Standard Epithelium Profiles in Transepithelial Photorefractive Keratectomy"

_optics, doi:10.3390/opt2040025_

Round 1

Reviewer 1 Report

In general, this article is innovative and has clinical value, though several minor errors need revision.

Line 16: It is better to add optimized before standard.

Line 51: You had better streamline the Materials and Methods part. As a matter of fact, the number of words in this part even exceeds that in the Discussion part.

Line 79: You mentioned 4 months here, but figure 4 shows 3m. Something went wrong.

Line 89: Given the diverse readership of Optics, it would be more advisable to provide a schematic diagram to illustrate the differences among A) the standard ablation profile provided by the software, B) the optimized standard ablation profile, and C) the customized ablation profile. That would be friendly to more readers.

Line 135:

  1. A) Can you provide the p value of each parameter?
  2. B) Did you use both eyes from the same subject, or preferably, one eye from the same subject?
  3. C) In the group A, were there any data pertaining to preoperative corneal epithelium thickness profile?I wonder whether the corneal epithelium thickness was comparable between the two groups prior to surgery.

Line 214: An error occurred. Figure 3Figure 7

Line 244: In this paragraph, you discussed about results from other papers, but did not mention about the similarity or difference between your study and other studies. You should discuss more details about the corneal epithelium thickness before and after surgery in group A and group B, if possible.

Line 277: You had better mention the limitations about your work. Firstly, the follow-up time was short, because the overall visual recovery is long for T-PRK, 4 months might not be enough to achieve a valid conclusion. Secondly, it is a retrospective study, so you might not be able to provide preoperative corneal epithelium thickness in the group A. There could be potential statistically significant difference between the two groups regarding the corneal epithelium thickness, hence an untrustworthy conclusion. 

Author Response

Comments and Suggestions for Authors

In general, this article is innovative and has clinical value, though several minor errors need revision.

Thank you!

Line 16: It is better to add optimized before standard.

Thank you! Added now. Abstract: „and compared themwith the results from a group treated with an optimizedstandard epithelium thickness

Line 51: You had better streamline the Materials and Methods part. As a matter of fact, the number of words in thispart even exceeds that in the Discussion part.

Thank you! Shortened now

Line 79: You mentioned 4 months here, but figure 4 shows 3m. Something went wrong.

Thank you! 4 months is correct, the label at figure 4 hasbeen corrected

Line 89: Given the diverse readership of Optics, it would bemore advisable to provide a schematic diagram to illustrate thedifferences among A) the standard ablation profile provided bythe softwareB) the optimized standard ablation profile, and C) the customized ablation profile. That would be friendlyto more readers.

Thank you! Actually, readers should be referred to ref 3: S. Arba Mosquera and S. T. Awwad, ‘Theoretical analysesof the refractive implications of transepithelial PRK abla-tions.’, Br J Ophthalmol, vol. 97, no. 7, pp. 905–911, Jul. 2013, doi: 10.1136/bjophthalmol-2012-302853; where theimplications and the rationale of the decision is fully disclosed.

We think that a diagram may be also provided in the form of a table, as follows:

Epithelial Profile

Central Thickness

PeripheralThickness

@4mm radial distance

Remarks

Standard

55µm

65µm

Population based epithelium covering>75% of the normal population

Optimized

(Standard + 5µm)

60µm

70µm

Optimized population based epitheliumcovering >95% of the normal population

Customized

(as measuredby OCT)

40-75µm

(as measuredcentrally usingMS-39)

Central thickness + 10µm

According to Impact of the Reference Point for Epithelial ThicknessMeasurements. Arba-Mosquera S, Awwad ST. J Refract Surg. 2020 Mar 1;36(3):200-207

Line 135:

1. A) Can you provide the p value of each parameter?

Thank you! Added now

2. B) Did you use both eyes from the same subject, orpreferably, one eye from the same subject?

Thank you! Excellent point.  We actually used both eyesfrom the same subject, yet for the statistical analyses the p-values were calculated considering the number of patientsand not the number of eyes.  This has been added now tosection 2.6.

3. C) In the group A, were there any data pertaining topreoperative corneal epithelium thickness profile?Iwonder whether the corneal epithelium thickness was comparable between the two groups prior to surgery.

 Thank you! Excellent point.  For group A, there was nosystematic data on preoperative corneal epitheliumthickness.  This group was performed before the MS-39 OCT (with the ability to provide epithelial maps) was available at the clinical centre.  We had previousexperiences (but not systematic data collections) of usingthe Optovue OCT to measured the central epithelium in specific patients.  We acknoowedge this now as a limitation, since strictly speaking we cannot excludedifferences between the corneal epithelium thickness forthe two groups prior to surgery.  Yet, at the light of theresults in group A and accepting that we were treatingnormal populations, webelievethat even if differencesbetween the corneal epithelium thickness for the twogroups prior to surgery may have existed, this would reachunlikely any meaningful clinical relevance.

This has been now added to the discussion:

For group A, there was no systematic data on preoperative corneal epithelium thickness.  This group was performed before the MS-39 OCT (with the ability toprovide epithelial maps) was available at the clinic.  Wehad previous experiences (but not systematic datacollections) of using the Optovue OCT to measured thecentral epithelium in specific pa-tients.  We acknoowedgethis as a limitation, since strictly speaking we cannotexclude differences between the corneal epitheliumthickness for the two groups prior to surgery.  Yet, at thelight of the results in group A and accepting that we weretreating normal populations, webelievethat even ifdifferences between the corneal epithelium thick-ness forthe two groups prior to surgery may have existed, thiswould reach unlikely any meaningful clinical relevance.

For the same reasons, we consider the report and analysisof postoperative epithelial thickness, which would only bepossible for group B, does not add further value to thiscomparison.  The specific analysis of the epithelial remodeling after TransPRK warrants further study.

Line 214: An error occurredFigure 3 → Figure 7

 Thank you! Corrected now!

Line 244: In this paragraph, you discussed about results fromother papers, but did not mention about the similarity ordifference between your study and other studies. You shoulddiscuss more details about the corneal epithelium thicknessbefore and after surgery in group A and group B, if possible.

 Thank you! Added now!

Line 277: You had better mention the limitations about yourwork. Firstly, the follow-up time was short, because theoverall visual recovery is long for T-PRK, 4 months might not be enough to achieve a valid conclusion. Secondly, it is a retrospective study, so you might not be able to providepreoperative corneal epithelium thickness in the group A. There could be potential statistically significant differencebetween the two groups regarding the corneal epitheliumthickness, hence an untrustworthy conclusion

 Thank you! Added now at the end of the disucssion:

Limitations of this work include the use of both eyes fromthe same patients (although statistical analyses have beenperformed based on the number of patients and not thenumber of eyes), the follow-up time was short (the overallvisual recovery is long for (transepithelial) PRK and 4 months might not be enough to achieve final stability; alt-hough the interest in this work was the comparisonbetween both groups at the 4-month follow-up and thefinal stability); it is a retrospective study and we are not able to provide corneal epithelium thickness for group A (there could be potential statistically significant differencebetween the two groups regarding the corneal epitheliumthickness, yet un-likely reaching any meaningful clinicalrelevance).

Reviewer 2 Report

The authors used anterior segment optical coherence (AS-OCT) to measure the epithelium thickness for further transepithelial photorefractive keratectomy and compared this method with a standard way. It is well organized and easy to read and follow. There are some places that need further clarification in description or revision.

  1. Can the authors show some AS-OCT images before and after TransPRK for the control group and the group treated with a customized epithelium thickness?
  2. For the figure in Line 213, the figure number has some problems.
  3. The abbreviation (CDVA) may not be directly used when it appeared in the abstract for the first time.

Author Response

Comments and Suggestions for Authors

The authors used anterior segment opticalcoherence (AS-OCT) to measure the epitheliumthickness for further transepithelial photorefractivekeratectomy and compared this method with a standard way. It is well organized and easy to readand follow. There are some places that needfurther clarification in description or revision.

 Thank you!

1. Can the authors show some AS-OCT imagesbefore and after TransPRK for the controlgroup and the group treated with a customizedepithelium thickness?

 Thank you! For group A, there was no systematicdata on preoperative corneal epithelium thickness.  This group was performed before the MS-39 OCT (with the ability to provide epithelial maps) was available at the clinic.  We had previous experiences(but not systematic data collections) of using theOptovue OCT to measured the central epithelium in specific pa-tients.  We acknoowedge this as a limitation, since strictly speaking we cannot excludedifferences between the corneal epithelium thicknessfor the two groups prior to surgery.  Yet, at the light of the results in group A and accepting that we weretreating normal populations, webelievethat even ifdifferences between the corneal epithelium thick-ness for the two groups prior to surgery may have existed, this would reach unlikely any meaningful clinicalrelevance.

For the same reasons, we consider the report and analysis of postoperative epithelial thickness, whichwould only be possible for group B, does not addfurther value to this comparison.  The specificanalysis of the epithelial remodeling after TransPRKwarrants further study.

This is included in the discussion

2. For the figure in Line 213, the figure numberhas some problems.

 Thank you! Corrected now!

3. The abbreviation (CDVA) may not be directlyused when it appeared in the abstract for thefirst time

 Thank you! Corrected now!

Round 2

Reviewer 2 Report

The authors answered all my questions and concerns.